# Investigation of Maternal Diet and *FADS1* Polymorphism Associated with Long-Chain Polyunsaturated Fatty Acid Compositions in Human Milk

**DOI:** 10.3390/nu14102160

**Published:** 2022-05-23

**Authors:** Sakurako Niwa, Terue Kawabata, Kumiko Shoji, Hiromitsu Ogata, Yasuo Kagawa, Kazuhiro Nakayama, Yoshiko Yanagisawa, Sadahiko Iwamoto, Nozomi Tatsuta, Kaname Asato, Takahiro Arima, Nobuo Yaegashi, Kunihiko Nakai

**Affiliations:** 1Faculty of Nutrition, Kagawa Nutrition University, 3-9-21 Chiyoda, Sakado 350-0288, Saitama, Japan; kawabata@eiyo.ac.jp (T.K.); shoji.kumiko@eiyo.ac.jp (K.S.); ogata.hiromitsu@eiyo.ac.jp (H.O.); kagawa@eiyo.ac.jp (Y.K.); 2Faculty of Home Economics, Gifu Women’s University, 80 Taromaru, Gifu City 501-2592, Gifu, Japan; 3Department of Integrated Biosciences, Graduate School of Frontier Sciences, The University of Tokyo, 5-1-5 Kashiwanoha, Kashiwa 277-8562, Chiba, Japan; knakayama@edu.k.u-tokyo.ac.jp; 4Division of Human Genetics, Center for Molecular Medicine, Jichi Medical University, 3311-1 Yakushiji, Shimotsuke 329-0498, Tochigi, Japan; yyanagi@jichi.ac.jp (Y.Y.); siwamoto@jichi.ac.jp (S.I.); 5Department of Development and Environmental Medicine, Tohoku University Graduate School of Medicine, 2-1 Seiryo-machi, Sendai 980-8575, Miyagi, Japan; nozomi@med.tohoku.ac.jp (N.T.); asatok@med.tohoku.ac.jp (K.A.); nakai-k@tokaigakuen-u.ac.jp (K.N.); 6Environment and Genome Research Center, Tohoku University Graduate School of Medicine, 2-1 Seiryo-machi, Sendai 980-8575, Miyagi, Japan; tarima@med.tohoku.ac.jp (T.A.); nobuo.yaegashi.c7@tohoku.ac.jp (N.Y.); 7School of Sport and Health Science, Tokai Gakuen University, Nishinohora 21-233, Miyoshi 470-0207, Aichi, Japan

**Keywords:** human milk, long-chain polyunsaturated fatty acids (LCPUFA), arachidonic acid (ARA), docosahexaenoic acid (DHA), maternal diet, *FADS1* SNP

## Abstract

Increasing the amount of long-chain polyunsaturated fatty acids (LCPUFA) in human milk is an important strategy for infant growth and development. We investigated the associations of LCPUFA compositions in human milk with maternal diet (especially fish and shellfish intake), with fatty acid Δ5 desaturase gene (*FADS1*) polymorphisms, and with gene-diet interactions. The present study was performed as part of an adjunct study of the Japan Environment and Children’s Study. The participants were 304 lactating females, who provided human milk 6–7 months after delivery. Fatty acids in human milk were analyzed by gas chromatography, and dietary surveys were conducted using a brief self-administered diet history questionnaire. We also analyzed a single nucleotide polymorphism of *FADS1* (rs174547, T/C). There was a significant difference in arachidonic acid (ARA) composition in human milk among the genotype groups, and the values were decreasing in the order of TT > TC > CC. The concentrations of eicosapentaenoic acid (EPA) and docosahexaenoic acid (DHA) were also different between TT and CC genotype, indicating a tendency for decreasing values in the same order. The composition of ARA showed significant gene–dietary interactions in multiple regression analysis, and the positive correlation between fish and shellfish intake and ARA composition in human milk was significant only in the CC genotype. Moreover, the factor most strongly associated with EPA and DHA composition in human milk was fish and shellfish intake. Therefore, it was suggested that increasing fish and shellfish intake in mothers may increase EPA and DHA composition in human milk, while increasing fish and shellfish intake in CC genotype mothers may lead to increased ARA composition in human milk.

## 1. Introduction

Arachidonic acid (ARA) and docosahexaenoic acid (DHA) are included in long-chain polyunsaturated fatty acids (LCPUFA) that are preferentially accumulated into the rapidly developing brain and retina during gestation and the first 2 years of infancy [1]. Therefore, ARA and DHA are considered important for the development of cognitive and visual functions [2,3,4]. Moreover, LCPUFA are compounds related to signal transduction [5] and affect immune function [6,7]. For infants who depend on breast feeding, increasing the amount of LCPUFA in human milk is an important strategy for infant growth and development. Therefore, we need to know the factors that determine the amount of LCPUFA in breast milk.

Fatty fish are a source of n-3LCPUFA such as EPA and DHA and are also rich in ARA. Significant positive correlations between fish intake and n-3LCPUFA composition in human milk were reported in observational studies in areas with both high fish intake [8] and low fish intake [9,10]. In another observational study of lactating women in the Philippines, it was reported that there was a quantitative response relationship between the weekly fish intake frequency of mothers and the DHA content in human milk [11]. A British intervention study in which 150 g of salmon was ingested twice a week from 20 weeks of gestation to childbirth also showed that the salmon intake group had significantly higher human milk DHA content than the control group [12]. Thus, many papers have demonstrated that consumed fish-derived fatty acids such as ARA and DHA increase the LCPUFA content in human milk.

Linoleic acid in the n-6 fatty acid group is converted to ARA, and α-linolenic acid in the n-3 fatty acid group is converted to EPA and DHA by desaturation and elongation [13]. The Δ 5 desaturase (D5D) and Δ 6 desaturase (D6D) that catalyze this desaturation reaction are encoded by *FADS1* and *FADS2* genes, respectively [14]. Up to the present, many single nucleotide polymorphisms (SNP) of these desaturase genes that are thought to be related to fatty acid metabolism and diseases have been reported [15,16]. In 2010, Nakayama et al. reported that the copy number of rs174547 C allele in the D5D gene *FADS1* was significantly associated with increased circulating TG in Japanese people [17]. After that, in the same SNP, we reported that the percentages of n-6 and n-3 LCPUFA in plasma or erythrocytes of the CC-genotype group were significantly lower compared with the TT-genotype group in the Japanese population [18,19]. Studies using stable isotope-labeled linoleic acid also found that participants with the CC genotype had significantly reduced production of the metabolite ARA compared to participants with the TT genotype [20]. Thus, in Japanese people, the *FADS1* SNP (rs174547) has a great effect on LCPUFA fatty acid formation from precursor fatty acids such as linoleic acid and α-linolenic acid.

The relationship between FADS polymorphisms in mothers and fatty acid composition in human milk is described in a 2020 systematic review by Conway et al. [21]. Here, they reviewed seven studies (2008–2017) on LCPUFA compositions in human milk and 24 FADS SNPs. Many of the seven studies suggested that FADS polymorphisms may affect LCPUFA composition in human milk. However, regarding Japanese people, there have not been reports on the relationships between FADS polymorphisms, including *FADS1* SNP (rs174547), and LCPUFA compositions in human milk.

As mentioned above, the determinants of LCPUFA composition in human milk include at least the factors of both the mother’s diet and gene polymorphisms. To our knowledge, studies reporting the interrelationships of the three factors—that is, LCPUFA composition of human milk, FADS polymorphisms, and n-3LCPUFA intakes in mothers, and moreover those that include gene-diet interactions—comprise only two reports, consisting of a Dutch cohort study [22] and a Taiwanese study [23]. In these two reports, mothers with genotypes with low LCPUFA synthesis did not have increased DHA content in human milk with increased intake of fish and shellfish, or DHA.

In the present study, we investigated the relationships of LCPUFA composition in human milk, with maternal diets, especially fish intake, and with maternal *FADS1* polymorphism (rs174547, T/C), as well as with gene–diet interactions. To achieve nutritional approaches aimed at lactating women for better delivery of infant nutrition, we believe it is of great significance to explore the factors that determine the LCPUFA composition in human milk.

## 2. Materials and Methods

### 2.1. Study Design 

In January 2011, the Ministry of the Environment started a nationwide birth cohort study, the Japan Environmental Children’s Study (JECS) [24]. The present study was performed as an adjunct study of the JECS, and the protocol and detailed outline of this adjunct study are explained elsewhere [25]. To answer the present study questions, we conducted a cross-sectional study for a subgroup of lactating women participating in the adjunct study.

The study protocol was approved by the Ethics Committee of the Tohoku University Graduate School of Medicine (No. 2010-322, 25 October 2010; No. 2010-421, 17 December 2010; No. 2012-1-154, 13 July 2012) and the Medical Ethics Committee of Kagawa Nutrition University (No. 187G, 18 January 2012; No. 269G, 10 January 2019). All participants agreed to participate voluntarily, then submitted written informed consent after consenting to the written explanation of the study purpose and the study details.

### 2.2. Participants

The study was conducted at the coastal area of Miyagi Prefecture. Pregnant women (first trimester to 12 weeks’ gestation) who visited participating hospitals for prenatal care were first recruited by the JECS. JECS participants in the coastal area were recruited by the adjunct study, and, finally, 1878 participants joined the adjunct study. From the adjunct study participants, those who had a single birth between July 2012 and August 2013, with breastfeeding at 6–7 months after delivery and with no fatty acid supplements were included in this study. Finally, 304 lactating females provided the breastmilk samples for analysis and examined in this study.

### 2.3. Human Milk Sampling

In accordance with the JECS manual, mothers were asked to collect approximately 30 mL of milk at home by hand squeezing without using a pump into a container with a lid. The milk was then immediately sent to the laboratory of Kagawa Nutrition University by refrigerated courier with ice packs. The human milk samples were stored at −80 °C until analysis.

### 2.4. Analysis of Fatty Acids in Human Milk 

The analysis method of Johannsson et al. [26], partially modified, was used to analyze the fatty acids in human milk. Briefly, toluene was added to thawed milk, and further individual fatty acids were methylated using methanol hydrochloride. The fatty acid methyl ester was extracted with hexane, and then gas-liquid chromatography analysis of the fatty acid methyl ester levels was conducted. The percentage of the peak area of each fatty acid to the total peak area of the measured fatty acids was calculated and expressed as the fatty acid composition (wt%).

### 2.5. Participants’ Basic Characteristics 

Data on basic characteristics including maternal age at delivery, maternal non-pregnant weight, height, pre-pregnancy body mass index (BMI) (kg/m^2^), parity, smoking and drinking status during pregnancy, maternal educational background, and infant characteristics including gestational age, birth weight, sex, and Apgar score were obtained from the JECS study (based on the jecs-ta-20190930 dataset released in September 2019).

### 2.6. Food Intake Questionnaire

Participants completed a brief self-administered diet history questionnaire (BDHQ) on the day of the examination for infants 6–7 months of age. Nutrient intakes that were obtained by the BDHQ were adjusted using the energy density method (i.e., the amount per 1000 kcal) and used in the analysis. The validity of the BDHQ was confirmed in a previous report using the 16-day weighted dietary record as the gold standard [27].

### 2.7. Genotyping

The genotyping method was described in a previous paper [19]. Briefly, a fully automated nucleic acid extraction system was used to extract DNA from whole blood. *FADS1* (rs174547, T/C) polymorphism was determined using the TaqMan Genotyping Assay System at the Collaborative Research Institute, Division of Human Genetics, Jichi Medical University.

### 2.8. Statistical Analysis

The normality of each variable was judged by the Shapiro–Wilk test and the normal probability plot, and if the data showed a non-normal distribution, the normality of the data was confirmed after logarithmic transformation and then used in the multiple regression analysis. To make comparisons according to genotype group, when the variable was continuous data, a Kruskal–Wallis test was conducted, followed by a post-hoc analysis to identify significant differences using the Steel–Dwass test; when the variable was categorical data, χ^2^ analysis was conducted. The relationships between two continuous variables were examined by Spearman’s rank correlation coefficient.

Multiple regression analysis was used to examine the association between LCPUFA composition in human milk and diet and gene polymorphisms in mothers, as well as gene-diet interactions. The objective variable was the LCPUFA composition in human milk, and the explanatory variables were the maternal genotype, the food group intakes (fish and shellfish, fats and oils, eggs) in mothers, and the gene-diet interaction term. The adjustment variables selected by the Stepwise method (variable increase/decrease method) were maternal age at delivery, pre-pregnancy BMI, parity, maternal smoking and passive smoking during pregnancy, maternal educational background, gestational age, infant sex, and birth season. Participants with missing gestational age data were excluded from the analysis (*n* = 6).

Significance was defined as *p* < 0.05. All statistical analyses were carried out using the JMP software package, version 16.0 (SAS Institute Inc., Cary, NC, USA).

## 3. Results

### 3.1. Characteristics of the Participants

The median age in all mothers at delivery was 31.2 years, the pre-pregnancy BMI was 21.0 kg/m^2^, the birth weight was 3101 g, and the height was 49.0 cm. Participant characteristics by *FADS1* rs174547 genotypes are shown in Table 1. Although there was a statistical association between parity and genotype (*p* < 0.05), there was no significant difference in other characteristics.

The gene polymorphisms of D5D involved in LCPUFA metabolism, rs174547 (T/C) in *FADS1*, were TT 37.5%, TC 49.7%, CC 12.8%, and the C allele frequency was 0.377. The genotype was shown to be in Hardy–Weinberg equilibrium by a χ^2^ test (*p* = 0.314).

### 3.2. LCPUFA Compositions in Human Milk

Table 2 indicates the LCPUFA compositions and its precursor fatty acid compositions of human milk by the TT, TC, and CC genotype of *FADS1*. It was shown that there were significant differences in γ-linolenic acid and ARA among the groups, with decreasing values in the order of TT > TC > CC. There were significant differences between TT and CC genotype in EPA and DHA, indicating a tendency toward decreasing values in the order of TT > TC > CC.

### 3.3. Relationships between Genotype, Maternal Diet and LCPUFA Compositions, and Gene–Diet Interaction 

Table 3 shows the correlation between food group intakes in mothers and LCPUFA compositions in human milk, stratified by *FADS1* genotype. Fish and shellfish intake in mothers was significantly positively correlated with ARA composition in human milk only in the CC genotype, and with EPA and DHA composition in the TT, TC, and CC genotypes. Egg intake in mothers was significantly positively correlated with ARA only in the TC genotype, and fats and oils intake were significantly negatively correlated with DHA only in the TT genotype. No significant correlation was found between meat intake in mothers and LCPUFA composition in human milk.

Table 4 shows the results of multiple regression analysis with LCPUFA compositions in human milk as the objective variable, and *FADS1* rs174547 genotypes and fish and shellfish, fats and oils, and egg intakes as explanatory variables. In model II, the gene–diet interactions were examined using the food groups in which the value of β was significant in model I. In model I, the β value of fish and shellfish intake was not significant for ARA in human milk; however, Table 3 indicates that ARA was significantly associated with fish and shellfish intake in the CC genotype. Therefore, we examined the gene–diet interaction with fish and shellfish in model II.

As a result of multiple regression analysis (model II), ARA in human milk was significantly correlated with genotype CC (vs. TT), fish and shellfish intake, egg intake, and interaction term: genotype CC (vs. TT) × fish and shellfish intake, of which genotype CC (vs. TT) showed the strongest correlation (β = −0.430; *p* < 0.001). EPA in human milk was significantly associated with genotype CC (vs. TT) and fish and shellfish intake, and the association with fish and shellfish intake was stronger (β = 0.472; *p* < 0.001) than the genotype. DHA in human milk was significantly associated with fish and shellfish intake and fat and oil intake, and the association of fish and shellfish intake was stronger (β = 0.505; *p* < 0.001) than fat and oil intake. No significant relationship was found between DHA in human milk and genotype.

## 4. Discussion

In the present study, we analyzed the LCPUFA composition in human milk of 6–7 months after delivery in 304 Japanese females and investigated the relationships of LCPUFA composition in human milk, with maternal diet, and with maternal gene polymorphisms, and, in addition, with gene-diet interactions. As a result, this study showed that ARA in human milk was related most strongly with genotype, but it was also related with fish and shellfish intake. Furthermore, a significant gene–diet interaction was observed in ARA. The factor most strongly associated with EPA and DHA in human milk was fish and shellfish intake in mothers.

A German cohort study [28] and a Chinese observational study [29] showed the relationships between the LCPUFA compositions in human milk and genotypes using the SNP (rs174547) of *FADS1*, the same used in the present study. In the German cohort study [28], LCPUFA compositions in human milk of 1.5 and 6 months after delivery were analyzed, and it was found that ARA alone in human milk of 6 months after delivery in a major allele homo (MM) group was significantly higher compared with a minor allele carrier (m) group. In the Chinese observational study [29], only ARA in human milk 22–25 days after delivery showed decreasing values in the order of MM > Mm > mm. In both studies, there were no significant differences between genotypes and n-3LCPUFA, such as EPA and DHA. There is a cohort study analyzing rs174561 in *FADS1*, rs174575 in *FADS2*, and rs3834458 in Intergenic [22], and a cohort study analyzing rs174553 in *FADS1*, rs174575 in *FADS2* [30]. In these studies, ARA, EPA, and DHA in human milk showed decreasing values in the order of MM > Mm > mm, but DHA composition was not significantly different for some of the SNPs. In addition, a Canadian cohort study [31] reported that only ARA in human milk 3–4 months after delivery showed decreasing values in the order of MM > Mm > mm in rs174556 of *FADS1* and rs174575 of *FADS2*.

In this study, we analyzed the genotype of rs174547 (T/C) in *FADS1* and found that ARA, EPA, and DHA compositions in human milk showed decreasing values in the order of TT > TC > CC—that is, MM > Mm > mm. These results were consistent with the results of the previous studies mentioned above [28,29,30,31]. As the number of the C allele of rs174547 in *FADS1* increases, the amount of synthesis or activity of the D5D encoded by *FADS1* decreases. As a result, the conversions from 20: 3n-6 to ARA in the n-6 series, and from 20: 4n-3 to EPA in the n-3 series, which are the reactions at the point of action of D5D, decline. Then, the synthesized quantities of ARA, EPA, and DHA downstream from the D5D action point decrease. Our study indicated that ARA, EPA, and DHA compositions in human milk showed decreasing values in the order of TT > TC > CC, so we considered that the differences in LCPUFA compositions in human milk by genotype directly reflect the differences in the synthesized quantities of LCPUFA.

Similar to ARA, there was a difference of MM > Mm > mm in the 18: 3n-6 composition in human milk, which was statistically significant. However, 18: 3n-6 is upstream from the point of action of the D5D enzyme and is not considered to be affected by D5D. Therefore, we assume that the participants with minor allele C of rs174547 in *FADS1* had reduced synthesis or activity of D6D, which converts linoleic acid to 18: 3n-6. The D5D and D6D genes are on the same chromosome, adjacent head-to-head [16], and the SNPs of these two genes are in linkage disequilibrium [17]. As a result, we believe that 18: 3n-6 showed the same significant difference as LCPUFA, even though it was located upstream of the D5D point of action.

The strength of the association to ARA composition in human milk was genotype > diet, and both EPA and DHA were diet > genotype. These results are consistent with the results of the previous cohort study [31]. Brenna et al. [32] reported that the coefficient of variation of DHA composition in human milk was 69%, whereas the coefficient of variation of ARA composition in human milk was 28%. It is assumed that the low coefficient of variation of ARA in human milk has been attributed to its high dependence on biosynthesis [33,34]. In the present study, a significant difference in gene–dietary interaction was observed only for ARA in human milk, and the positive correlation between fish and shellfish intake and ARA composition in human milk by genotype was significant only in the minor allele homozygous CC participants (Table 3). As mentioned above, participants with minor allele homozygous CC may have reduced conversion of linoleic acid to ARA. For the milk of CC genotype participants, even ARA composition, which is highly dependent on biosynthesis, was considered to be more strongly related to diet (fish and shellfish intake) than the milk of other genotypes. 

Many observational studies [8,9,10,11] and intervention studies [12] have shown a significant positive correlation between fish and shellfish intake and n-3LCPUFA content in human milk. A dietary 13C-labeled fatty acids study showed that supplemented labeled DHA was detected at a peak in human milk after 6–12 hours, clarifying that DHA content in human milk was influenced by the most recent dietary intake [35]. The authors demonstrated a linear dose–response relationship between DHA intake and DHA content in human milk. Previous studies demonstrated that n-3LCPUFA content in human milk tended to show no significant difference among FADS genotypes groups compared with n-6LCPUFA [28,29,31]. Therefore, we considered that the determinant of n-3LCPUFA content in human milk was more strongly related to diet than genotype.

When linoleic acid and α-linolenic acid with 18 carbon atoms derived from vegetable oils are metabolized into ARA and DHA, respectively, the n-6 series and n-3 series use the same enzyme [16]. N-6 and n-3 metabolisms are considered to be in a competitive relationship [13]. When we consume common oils such as rapeseed oil and soybean oil, we usually have a higher intake of linoleic acid than α-linolenic acid. If linoleic acid is increased in the body, LCPUFA metabolizing enzyme will be used for n-6PUFA metabolism, leading to a relative decrease in the degree of metabolism of n-3PUFA, and the production of the metabolite DHA will thus decrease. Moreover, in a study in which corn oil rich in linoleic acid was given to lactating rats, it was reported that the rats with high intake of corn oil expressed less D5D and D6D in the liver than the rats with low intake of corn oil [36,37]. From the above, it was thought that an increase in fat and oils intake, that is, an increase in linoleic acid intake, may lead to inhibition of n-3LCPUFA metabolism and decrease the amount of biosynthesis of DHA, which is the product of n-3 series metabolism. In our present study, there was a significant negative association between DHA content in human milk and fat and oils intake, and we speculate that the intakes of linoleic acid in fats and oils could have been a major factor in producing this result.

This study has some limitations. First, we did not investigate human milk intake, so it was not possible to estimate the absolute amount of fatty acids actually ingested by infants. Therefore, we did not refer to the relationship between fatty acids in human milk and infant growth. Second, we did not have information on the participants’ most recent dietary intake before milk collection. A tracer study [35] has shown that DHA content in human milk reflects the most recent dietary intake. If we could obtain information on the most recent dietary intake prior to milk collection, we believe that the relationship between diet in mothers and DHA in human milk could be analyzed more accurately.

## 5. Conclusions

ARA composition in human milk showed significant gene–dietary interactions, and the positive correlation between fish and shellfish intake and ARA in human milk was significant only in the CC genotype. That is, in those with minor allele homozygous CC, ARA composition in the milk may be more strongly influenced by dietary intake than in those with other genotypes. Although EPA and DHA in human milk are associated with genotype, it was found that the strongest factor determining their content in human milk was fish and shellfish intake. The above considerations suggest that increasing fish and shellfish intake in mothers may increase EPA and DHA content in human milk, and increasing the fish and shellfish intake of CC genotype mothers may lead to increased ARA content in human milk.

## Figures and Tables

**Table 1 nutrients-14-02160-t001:** Participant characteristics by *FADS1* rs174547 genotypes.

	*FADS1*; rs174547 Genotype						*p* *
	TT (*n* = 114)		TC (*n* = 151)		CC (*n* = 39)		
Mothers							
Age at delivery, y	32.2	(29.2−35.0)	31.0	(28.0−34.4)	31.9	(28.0−37.3)	0.135
Non-pregnant physique							
Height, cm	158	(155−162)	158	(155−162)	158	(154−161)	0.654
Weight, kg	52	(48−58)	53	(49−58)	50	(47−57)	0.278
BMI, kg/m^2^	21.2	(19.1−22.8)	20.8	(19.5−22.9)	20.2	(18.6−23.0)	0.364
Parity							**0.048**
1	82	(73.2%)	83	(56.5%)	22	(56.4%)	
≳2	30	(26.8%)	64	(43.5%)	17	(43.6%)	
Smoking during pregnancy ^†^							0.911
Smoker	21	(18.4%)	27	(17.9%)	6	(15.4%)	
Non-smoker	93	(81.6%)	124	(82.1%)	33	(84.6%)	
Passive smoking during pregnancy ^‡^							0.648
Passive smoker	68	(59.6%)	97	(64.2%)	26	(66.7%)	
Non-passive smoker	46	(40.4%)	54	(35.8%)	13	(33.3%)	
Mothers’ educational background							0.059
Middle school/High school	52	(45.6%)	60	(39.7%)	13	(33.3%)	
Technical college/Junior college/ Vocational school	53	(46.5%)	64	(42.4%)	16	(41.0%)	
University/Graduate school	9	(7.9%)	27	(17.9%)	10	(25.6%)	
Intake of food groups							
Fish and shellfish, g/1000 kcal	33.9	(21.9−47.2)	31.0	(21.8−40.6)	28.0	(19.5−42.5)	0.178
Meat, g/1000 kcal	32.9	(25.4−44.0)	36.2	(28.3−46.9)	36.9	(28.4−45.2)	0.204
Eggs, g/1000 kcal	17.0	(12.7−26.4)	18.4	(12.6−28.1)	15.8	(10.3−26.3)	0.508
Fats and oils, g/1000 kcal	4.7	(3.6−6.4)	5.2	(4.1−7.0)	5.2	(4.4−8.4)	0.100
Infants							
Gestational age, d	276	(269−281)	277	(270−282)	276	(269−284)	0.648
Sex							0.768
Male	61	(53.5%)	74	(49.0%)	20	(51.3%)	
Female	53	(46.5%)	77	(51.0%)	19	(48.7%)	
Season at birth							0.683
Spring, March–May	43	(37.7%)	51	(33.8%)	9	(23.1%)	
Summer, June–August	33	(29.0%)	49	(32.5%)	17	(43.6%)	
Autumn, September-November	12	(10.5%)	14	(9.3%)	4	(10.3%)	
Winter, December-February	26	(22.8%)	37	(24.5%)	9	(23.1%)	

Median (25th percentile–75th percentile) or *n* (%). * Kruskal–Wallis test was used for continuous variables, and χ^2^ test was used for categorical variables. Value in bold is statistically significant. ^†^ Subjects who answered “smoked during pregnancy” or “previously smoked, but stopped due to this pregnancy” in one or both of the first and second trimesters of pregnancy, were classified as smokers. ^‡^ In one or both of the first and second trimesters of pregnancy, participants who answered that they were exposed to second-hand cigarette smoke from others at home, at work, or indoors “every day”, “4–6 days a week”, “2–3 days a week”, or “about once a week” were classified as passive smokers.

**Table 2 nutrients-14-02160-t002:** Fatty acid compositions in human milk by *FADS1* rs174547 genotypes.

		*FADS1*; rs174547 Genotype						Kruskal–Wallis Test
		TT (*n* = 114)		TC (*n* = 151)		CC (*n* = 39)		*p* *
Total SFA, %		40.6	(38.4–43.3)	40.2	(36.6–43.9)	40.8	(38.6–44.3)	0.449
C16:0	Palmitic acid	21.3	(20.1–22.4)	21.1	(19.5–22.8)	21.4	(20.2–22.6)	0.461
Total MUFA, %		41.2	(39.6–43.7)	41.7	(39.3–43.9)	41.6	(40.0–43.5)	0.817
C18:1	Oleic acid ^†^	38.5	(36.9–40.7)	39.0	(36.8–41.1)	38.8	(37.6–40.6)	0.670
Total PUFA, %		17.5	(15.5–19.4)	18.0	(16.1–20.0)	17.0	(14.7–18.3)	0.106
Total n-6 PUFA, %		15.0	(13.4–16.2)	15.3	(13.7–16.9)	14.5	(13.3–15.5)	0.080
C18:2n-6	Linoleic acid	13.8	(12.4–15.1)	14.4	(12.8–16.0)	13.8	(12.4–14.7)	0.070
C18:3n-6	GLA	0.12	(0.10–0.15) ^a^	0.09	(0.08–0.11) ^b^	0.07	(0.05–0.09) ^c^	**<0.001**
C20:4n-6	ARA	0.42	(0.36–0.46) ^a^	0.34	(0.30–0.39) ^b^	0.31	(0.26–0.34) ^c^	**<0.001**
Total n-3 PUFA, %		2.54	(2.08–3.07)	2.50	(2.01–3.11)	2.34	(1.89–2.93)	0.482
C18:3n-3	α-linolenic acid	1.37	(1.03–1.65)	1.45	(1.15–1.90)	1.44	(1.12–1.63)	0.133
C20:5n-3	EPA	0.16	(0.10–0.27) ^a^	0.13	(0.08–0.22) ^ab^	0.10	(0.06–0.19) ^b^	**0.002**
C22:6n-3	DHA	0.61	(0.42–0.87) ^a^	0.51	(0.36–0.76) ^ab^	0.45	(0.34–0.69) ^b^	**0.018**

Median (25th percentile–75th percentile). SFA, saturated fatty acids; MUFA, monounsaturated fatty acids; PUFA, polyunsaturated fatty acids; GLA, γ-linolenic acid; ARA, arachidonic acid; EPA, eicosapentaenoic acid; DHA, docosahexaenoic acid. * Kruskal–Wallis test was conducted, followed by a post-hoc analysis to identify significant differences using the Steel–Dwass test. Values within a row with unlike superscript letters were significantly different (*p* < 0.05). Values in bold are statistically significant. ^†^ Total value of C18: 1n-9 and C18: 1n-7.

**Table 3 nutrients-14-02160-t003:** Correlation between food group intakes in mothers and LCPUFA compositions in human milk by *FADS1* rs174547 genotypes.

	Fatty Acids in Human Milk, wt%					
Food Group Intakes, g/1000 kcal	C20:4n-6 (ARA)		C20:5n-3 (EPA)		C22:6n-3 (DHA)	
*FADS1*; rs174547 Genotype	rs * (*p*)		rs * (*p*)		rs * (*p*)	
Fish and shellfish						
TT *n* = 114	−0.037	(0.693)	**0.253**	**(0.007)**	**0.281**	**(0.002)**
TC *n* = 151	0.122	(0.137)	**0.405**	**(<0.001)**	**0.400**	**(<0.001)**
CC *n* = 39	**0.446**	**(0.004)**	**0.508**	**(0.001)**	**0.633**	**(<0.001)**
Meat						
TT *n* = 114	−0.052	(0.580)	−0.053	(0.575)	−0.128	(0.175)
TC *n* = 151	0.150	(0.066)	−0.007	(0.936)	−0.020	(0.806)
CC *n* = 39	−0.056	(0.734)	−0.159	(0.333)	−0.152	(0.356)
Eggs						
TT *n* = 114	0.115	(0.222)	0.009	(0.921)	−0.011	(0.907)
TC *n* = 151	**0.252**	**(0.002)**	−0.014	(0.867)	0.032	(0.696)
CC *n* = 39	0.124	(0.454)	−0.086	(0.601)	−0.037	(0.822)
Fats and oils						
TT *n* = 114	−0.072	(0.446)	−0.112	(0.234)	**−0.186**	**(0.048)**
TC *n* = 151	0.079	(0.333)	−0.135	(0.099)	−0.102	(0.212)
CC *n* = 39	−0.156	(0.342)	0.058	(0.726)	−0.030	(0.855)

* Spearman’s rank correlation coefficient was conducted. Values in bold are statistically significant.

**Table 4 nutrients-14-02160-t004:** The results of multiple regression analysis with LCPUFA compositions in human milk as the objective variable, and *FADS1* rs174547 genotypes and fish and shellfish, fats and oils, and egg intakes as explanatory variables.

					Fatty Acids in Human Milk, wt%							
	C20:4n-6 (ARA)				C20:5n-3 (EPA) ^‡^				C22:6n-3 (DHA) ^‡^			
	Multiple Regression Analysis				Multiple Regression Analysis				Multiple Regression Analysis			
	Ⅰ		Ⅱ		Ⅰ		Ⅱ		Ⅰ		Ⅱ	
	β * (*p*)		β * (*p*)		β * (*p*)		β * (*p*)		β * (*p*)		β * (*p*)	
*FADS1*; rs174547 genotype ^†^												
TC	−0.110	(−0.092)	−0.115	(0.078)	0.055	(0.417)	0.068	(0.320)	−0.002	(0.982)	0.008	(0.906)
CC	**−0.439**	**(<0.001)**	**−0.430**	**(<0.001)**	**−0.178**	**(0.010)**	**−0.196**	**(0.006)**	−0.078	(0.254)	−0.087	(0.216)
Intakes, g/1000 kcal												
Fish and shellfish ^‡^	0.064	(0.211)	**0.163**	**(0.021)**	**0.413**	**(<0.001)**	**0.472**	**(<0.001)**	**0.432**	**(<0.001)**	**0.505**	**(<0.001)**
Fats and oils	−0.059	(0.245)	−0.057	(0.257)	**−0.134**	**(0.011)**	−0.100	(0.072)	**−0.144**	**(0.006)**	**−0.119**	**(0.031)**
Eggs ^‡^	**0.175**	**(0.001)**	**0.140**	**(0.016)**	−0.052	(0.321)	−0.050	(0.348)	−0.034	(0.521)	−0.033	(0.533)
Interaction terms ^†^												
TC × Fish and shellfish ^‡^			−0.087	(0.271)			−0.005	(0.948)			−0.026	(0.746)
CC × Fish and shellfish ^‡^			**0.192**	**(0.038)**			0.114	(0.227)			0.140	(0.138)
TC × Fats and oils							−0.085	(0.167)			−0.043	(0.480)
CC × Fats and oils							0.115	(0.063)			0.098	(0.111)
TC × Eggs ^‡^			0.050	(0.450)								
CC × Eggs ^‡^			−0.077	(0.251)								
Adjusted R-squared	R^2^ = 0.304		R^2^ = 0.308		R^2^ = 0.243		R^2^ = 0.251		R^2^ = 0.248		R^2^ = 0.254	
Model significance (*p*)	<0.001		<0.001		<0.001		<0.001		<0.001		<0.001	

*n* = 298. Analysis was conducted by excluding subjects with missing parity data. * Standardized partial regression coefficient. Values in bold are statistically significant. ^†^ Using the value of TT as the reference. ^‡^ Data were used in statistical analysis after logarithmic transformation. Both Model I and II were adjusted maternal age at delivery, BMI during non-pregnancy, parity, maternal smoking and passive smoking during pregnancy, maternal educational background, gestational age, infant sex, and birth season.

## Data Availability

Not applicable.

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
