# Peer review of "Investigation of Maternal Diet and FADS1 Polymorphism Associated with Long-Chain Polyunsaturated Fatty Acid Compositions in Human Milk"

_nutrients, 2022, doi:10.3390/nu14102160_

Round 1

Reviewer 1 Report

The authors try to investigate the relationships of LCPUFA composition in 100 human milk, with maternal diets, especially fish intake, and with maternal FADS1 poly- 101 morphism (rs174547, T/C), as well as with gene-diet interactions.

It is an interesting article. While some concerns are followings.

1.          In 2.7. Can you show the normality data of your judgments?

2.          Table 1. The height of the TT, TC, CC groups all are 158cm? Especial in the TT and TC groups both are 158 (155-162) cm? Can you show the original data?

3.          Table 1. Gestational age. Can you show the original data?

4.          Table 3. There is a typo in the “Eggs” to be fats and oils?

Author Response

  1. In 2.7. Can you show the normality data of your judgments?

→ Thank you for your comment. The normality of the data was relevant in the multiple regression analysis shown in Table 4. I did a log transformation when the data were non-normal; I have rewritten line 171. The variables not log-transformed in Table 4, which are arachidonic acid composition in human milk and fats and oils intake, are the data determined to be normal. The other analyses are essentially used nonparametric tests. The footnote in Table 1, " Analysis of variance was used for continuous variables of normal distribution," was my mistake and has been deleted. The data for which normality was found in Table 1 are age at delivery, height and fats and oils intake, and analysis of variance did not change the results.

In line 168, "Statistical Analysis" was corrected from 2.7. to 2.8.

  1. Table 1. The height of the TT, TC, CC groups all are 158cm? Especial in the TT and TC groups both are 158(155-162) cm? Can you show the original data?

→ Thank you for your comment. We have checked the original data. For maternal height, TT and TC had the same median, same 25th percentile, and 75th percentile values. Mean values differed between TT and TC, with TT at 158.1 cm and TC at 158.5 cm.

  1. Table 1. Gestational age. Can you show the original data?

→ Thank you for your comment. We have also checked the original data for the gestational age in Table 1 and found no errors.

  1. Table 3. There is a typo in the “Eggs” to be fats and oils?

→ Thank you for the correction. Table 3 has been corrected.

Reviewer 2 Report

I have carefully with great interest read the manuscript entitled: Investigation of maternal diet and FADS1 polymorphism associated with long-chain polyunsaturated fatty acid compositions in human milk.
The aim of the present work was to analyze the LCPUFA composition in human milk of 6-7 months after delivery in 304 Japanese females, and investigated the relationships of LCPUFA composition in human milk, with maternal diet, and with maternal gene polymorphisms, and gene-diet interactions. It is really interesting to find a relationship between genes and diet and mothers' breast milk composition.
I appreciate the really huge authors' work and the multiplicity of using methods: questionnaire,gas-liquid chromatography, genotyping,  
The manuscript is interesting and well written. References are correctly presented. 

Author Response

Thank you for your comments.